# Regularized Data Programming with Automated Bayesian Prior Selection

## Abstract

The cost of manual data labeling can be a significant obstacle in supervised learning. Data programming (DP) offers a weakly supervised solution for training dataset creation, wherein the outputs of user-defined programmatic labeling functions (LFs) are reconciled through unsupervised learning. However, DP can fail to outperform an unweighted majority vote in some scenarios, including low-data contexts. This work introduces a Bayesian extension of classical DP that mitigates failures of unsupervised learning by augmenting the DP objective with regularization terms. Regularized learning is achieved through maximum a posteriori estimation with informative priors. Majority vote is proposed as a proxy signal for automated prior parameter selection. Results suggest that regularized DP improves performance relative to maximum likelihood and majority voting, confers greater interpretability, and bolsters performance in low-data regimes.

## 1 Introduction

Data programming (DP) is a paradigm for training dataset creation wherein weakly supervised label generation is encoded by user-defined labeling functions (LFs) (Ratner et al., 2016; 2017). Automated data labeling with programmatic weak supervision offers a scalable alternative to manual labeling, whose costliness is a central challenge in machine learning (ML) (Liang et al., 2022). The weak supervision signals offered by LFs allow for inexpensive yet noisy label generation, with LFs ranging from simple keyword lookups to wrappers for pre-trained language models (Zhang et al., 2022). Though each LF typically labels only a subset of observations, a single observation can receive labels from multiple LFs (Figures 1, A.1). As individual LFs range in quality and their overlapping labels can conflict, the automatic denoising process must account for the unknown accuracy rate of each LF. Ratner et al. (2016) cast DP as a generative model that reconciles cheap, noisy, and conflicting LF outputs by learning these accuracy rates while treating ground truth labels as latent variables. The resulting labeled data can be used for discriminative model training, information extraction tasks, or knowledge base creation (Ratner et al., 2017; Kuleshov et al., 2019).

The utility of DP for research and industrial applications (Bach et al., 2019; Bringer et al., 2019) has motivated several adaptations, including adversarial DP (Arachie & Huang, 2019), interactive weak supervision (Boecking et al., 2021; Hsieh et al., 2022), and semi-supervised DP (Maheshwari et al., 2021). While conventional DP handles discrete labels, extensions have been proposed for LFs that output continuous values (Chatterjee et al., 2020) or individual loss functions that are aggregated to train a neural network (Sam & Kolter, 2023). Integration with large language models has also seen recent success (Arora et al., 2022; Smith et al., 2022; Lang et al., 2022a). However, the DP generative model can fail to produce accurate labels for some tasks. The state-of-the-art Snorkel DP model (Ratner et al., 2017) has been seen to underperform relative to an unweighted majority vote of LFs (MV) on some benchmarks (Sam & Kolter, 2023). Label quality can suffer when unlabeled training data are scarce, with generative model performance improving as training set size increases (Sam & Kolter, 2023; Dunnmon et al., 2020).

We seek to mitigate these failure modes of classical DP by incorporating informative priors, as Bayesian priors serve as strong regularizers under limited data. This method regularizes unsupervised learning by specifying user beliefs over LF accuracies, preventing underperformance relative to MV and resulting in

higher quality labels in low-data regimes. The proposed method incorporates previously under-exploited signals from MV at both training and inference time, providing an intuitive Bayesian alternative to previous attempts at regularizing DP (Chatterjee et al., 2020).

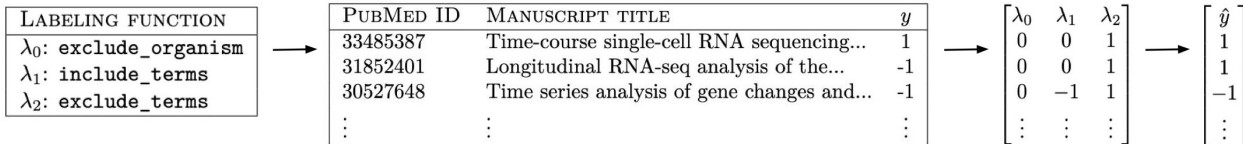

Figure 1: Automatic data labeling with programmatic weak supervision. This schematic illustrates the manuscript title filtering pipeline that motives this work. From left to right: 1) The user constructs LFs based on their domain expertise. 2) LFs are applied to textual data, whose ground truth labels $y$ are unavailabe for model training but may be available for inference on test data. 3) The sparse LF output matrix encodes the votes of each LF per observation, where 1 denotes a positive label, -1 denotes a negative label, and 0 denotes abstention. 4) The trained DP model automatically denoises overlapping and conflicting LF labels to yield $\hat{y}$, the predicted label vector.

## 1.1 Motivating Application: Biomedical Information Extraction

Manual data labeling is a key challenge limiting the adoption of ML in biomedical literature mining (Shin et al., 2015), driving reliance on less sophisticated approaches (Hofmann & Klinkenberg, 2016). Labeling complex scientific data requires significant domain expertise (Zhang, 2015), reducing the potential for crowd-sourcing (Dalvi et al., 2013; Zhang et al., 2014) and motivating interest in inexpensive, automated solutions. DP has facilitated extraction of protein–protein interactions from full-texts (Mallory et al., 2015), curation of a genome-wide association study knowledge base (Kuleshov et al., 2019), developement of a chemical reaction database (Mallory et al., 2020), and creation of a MRI dataset that outperformed a hand-labeled counterpart (Fries et al., 2019). Cross-modal weak supervision has been used to generate synthetic labels over textual data for training ML models over clinical imaging data, saving significant time in expert data labeling (Dunnmon et al., 2020).

The present work is motivated by the potential for regularized DP to support biomedical manuscript curation for large-scale literature reviews. Meta-analyses and systamatic reviews generally require human experts to filter hundreds or thousands of manuscripts for relevancy, prompting interest in ML-facilitated screening methods (Van De Schoot et al., 2021). To simulate the use of DP for this task, we introduce an original LF dataset of biomedical manuscript titles (Figure 1, Table A.1).

## 1.2 Contributions

This work introduces 1) a regularized DP objective that mitigates failure modes of classical DP in low-data regimes, 2) an automated prior selection procedure, and 3) a new DP benchmark dataset. The proposed Bayesian extension enables users to specify prior beliefs over LF accuracies and unknown ground truth labels by adapting the DP objective from maximum likelihood estimation (MLE) to maximum a posteriori estimation (MAP). To simplify the principled selection of prior parameters, an automated process treats an unweighted MV of LFs over the training data as a proxy for ground truth. To explore the use of regularized DP for biomedical literature curation, a novel LF dataset that simulates curation of a transcriptomics systematic review is released as a DP benchmark under the Open Data Commons. Data, source code, and experiments are available on GitHub.[1]

## 2 Methodology

The DP framework generates synthetic labels for a dataset $\{x_i, y_i\}_{i=1}^n$ whose true labels $y_i \in \{-1, 1\}$ are unknown. Synthetic labels $\hat{y}_i$ are derived from noisy LFs $\{\lambda_j\}_{j=1}^m$, where each LF $\lambda_j$ is characterized by its

---

[1]https://github.com/regularized-dp/regularized-data-programming/

coverage $\beta_j$ (the probability that the LF votes rather than abstains) and its accuracy $\alpha_j$ (the probability that the LF votes correctly). All mathematical notation is summarized in Table 1.

| VARIABLE | MEANING |
|---|---|
| $(x, y) \in \mathcal{X} \times \{-1, 1\}$ | Observations and unknown labels. |
| $m, n$ | Total LFs, total observations. |
| $i \in [1..n]$ | Index over $x$, $y$, and $\hat{y}$. |
| $j \in [1..m]$ | Index over $\lambda, \alpha$, and $\beta$. |
| $\Lambda$ | A $n \times m$ LF output matrix. |
| $\lambda_i : \mathcal{X} \mapsto \{-1, 0, 1\}^m$ | Outputs of every LF for a given $x_i$. |
| $\lambda_j : \mathcal{X} \mapsto \{-1, 0, 1\}^n$ | Outputs of a given LF for every $x_i$. |
| $\lambda_{ij} \in \{-1, 0, 1\}$ | Output of LF $\lambda_j$ for $x_i$; $0 =$ abstain. |
| $\hat{y}$ | Predicted label vector given $\Lambda$. |
| $\hat{y}_i$ | Predicted label for $x_i$ given $\lambda_i$. |
| $\alpha \in \mathbb{R}^m$ | Vector of all LF accuracies. |
| $\beta \in \mathbb{R}^m$ | Vector of all LF coverages. |
| $(u_j, v_j)$ | Beta distribution parameters. |
| $p$ | Bernoulli distribution parameter. |

Table 1: Variables referenced in model definition.

## 2.1 Model Definition

DP infers the unknown true labels $y$ from LF outputs by specifying a probabilistic model $\mathbf{P}(\lambda_{ij}, y_i \mid \alpha_j, \beta_j)$, in which $y_i$ is a latent variable and $\mathbf{P}(\lambda_{ij}|y_i, \alpha_j, \beta_j) =$

$$\begin{cases} 1 - \beta_j & \text{if } \lambda_{ij} = 0 \\ \alpha_j \beta_j & \text{if } \lambda_{ij} = y_i \\ (1 - \alpha_j)\beta_j & \text{if } \lambda_{ij} = -y_i \end{cases} \tag{1}$$

where $\lambda_{ij} = 0$ denotes abstention by $\lambda_j$.

**Bayesian formulation** We introduce priors over $y_i$ and $\alpha_j$ as $\mathbf{Ber}(y_i; p)$ and $\mathbf{Beta}(\alpha_j; u_j, v_j)$, respectively. The choice of a beta prior is motivated by its conjugacy with the Bernoulli likelihood of the DP model. This yields a Bayesian latent variable model of the form

$$\prod_{i=1}^{n} \mathbf{P}(\lambda_i, \alpha, \beta) = \prod_{i=1}^{n} \big[ \mathbf{P}(\lambda_i, y_i = 1, \alpha, \beta) \tag{2}$$
$$+ \quad \mathbf{P}(\lambda_i, y_i = -1, \alpha, \beta) \big]$$

such that $\mathbf{P}(\lambda_i, y_i, \alpha, \beta)$

$$= \mathbf{P}(\lambda_i|y_i, \alpha, \beta)\mathbf{P}(y_i)\mathbf{P}(\alpha)\mathbf{P}(\beta) \tag{3}$$

$$= \prod_{j=1}^{m} \mathbf{P}(\lambda_{ij}|y_i, \alpha_j, \beta_j)\mathbf{P}(y_i)\mathbf{P}(\alpha_j)\mathbf{P}(\beta_j). \tag{4}$$

## 2.2 Learning and Inference

**MAP estimation** In classical DP (Ratner et al., 2016), model parameters are learned via MLE: $\arg\max_{\alpha,\beta} \mathbf{P}(\Lambda|\alpha, \beta)$. The proposed method instead computes MAP estimates

$$\hat{\alpha}, \hat{\beta} = \arg\max_{\alpha,\beta} \mathbf{P}(\Lambda, \alpha, \beta). \tag{5}$$

In practice, $\hat{\alpha}$ is estimated through stochastic gradient descent (SGD) and $\hat{\beta}$ is fixed as the observed coverage rate on the training set. Alternatively, $\hat{\beta}$ can be treated as a learned parameter with a beta conjugate prior analogous to $\hat{\alpha}$. An experimental comparison found that learning $\hat{\beta}$ does not improve results on our data, as described in Table A.2. Thus, we opted for model simplicity by setting a fixed $\hat{\beta}$ for further experiments.

**Inference**  Observations are assigned the most likely label under the Bayesian model:

$$\hat{y} = \arg\max_y \mathbf{P}(\Lambda, y, \hat{\alpha}, \hat{\beta}). \tag{6}$$

Abstention tie-breaking enables the model to assign $\hat{y}_i = 0$ when neither label is more probable under the Bayesian model. This allows for the possibility of incomplete prediction coverage in $\hat{y}$.

## 2.3  Prior Selection and Model Regularization

Regularized learning is motivated by the observation that MV can outperform MLE when training data are scarce. While priors can be informed by user beliefs, we propose strategies for automatically selecting priors over $y$ and $\alpha$ that exploit $\hat{y}_{mv}$, the labels predicted by MV over the training data.

**Priors over $y$**  A single parameter $p$ is applied to all Bernoulli priors over $y$. Wherever $\hat{y}_{mv}$ votes, we set $p \geq 0.5$ such that sufficiently strong priors ($p$ approaching 1) will force the model to recapitulate every vote in $\hat{y}_{mv}$. While abstentions in $\hat{y}_{mv}$ are assigned an uninformative prior, the model can be forced to abstain on instances where MV abstains. This option exploits the observation that abstention forcing can improve performance on datasets where MV outperforms MLE, as demonstrated experimentally. The value of $p$ is tuned via grid search on the validation set, along with the boolean flag for forced abstention.

**Priors over $\alpha$**  Priors over $\alpha_j$ are beta distributions with parameters $(u_j, v_j)$ whose means approximate LF accuracies. Prior strength is dictated by the magnitude of parameters $(u_j, v_j)$ and is selected through grid search on validation data. Beta distribution means can be automatically selected by computing the accuracy of each $\lambda_j$ with respect to $\hat{y}_{mv}$, as a proxy for ground truth. We refer to models employing this heuristic as $\text{MAP}_{mv}$. This heuristic obviates human expertise and is cheaper than random search, but performant results are not guaranteed. As an experimental control, a theoretical upper bound on prior quality was obtained by setting distribution means to $\alpha^*$, the LF empirical accuracies with respect to the training set. These simulated optimal priors provide a benchmark for automated prior selection, termed $\text{MAP}_{\alpha^*}$.

## 3  Experimental Design

Experimental objectives are to demonstrate impacts of regularization on 1) the added value of unsupervised learning relative to MV, 2) performance in low-data contexts, and 3) the ability to infer LF accuracies. All experiments used abstention tie-breaking to assess variation in $\hat{y}$ coverage. Datasets were randomly split into training (80%) and testing (20%) sets, with 10% of training data held out as a validation set. Baseline models underwent grid search hyperparameter tuning with evaluation on validation data, as described in Table A.3.

### 3.1  Baseline Models

**Maximum likelihood model**  This model implements the objective defined by Ratner et al. (2016) in PyTorch. The MAP model is implemented identically, with the addition of priors over $\alpha$ and $y$.

**Majority vote model**  Unweighted MV offers a naive baseline against which to compare trained models. This model assigns each observation to the class that was selected by the majority of LFs for that data point.

**Snorkel labeling model**  Snorkel[2] (Ratner et al., 2017) is a DP framework with demonstrated utility at industrial scale (Bach et al., 2019). This baseline serves as an example of state-of-the-art DP.

---

[2]https://www.snorkel.org/

**CAGE** This baseline offers a non-Bayesian DP regularization method for continuous and discrete LFs, with *quality guides* representing user estimates of LF accuracies (Chatterjee et al., 2020). The CAGE likelihood for discrete LFs is employed, which differs from Ratner et al. (2016) by introducing $k$ parameters per LF (where $k$ is the total levels of $y$). This method requires all LFs to vote only in one direction (1 or -1), precluding its use for the transcriptomics dataset presented in this work.

**Support vector machine** A regularized linear support vector machine (SVM) with a hinge loss optimized through SGD serves as a supervised baseline. SVMs were implemented in scikit-learn using bag-of-words features. Unlike the DP models, this baseline has access to training labels.

### 3.2 Datasets

**TubeSpam** This dataset (Alberto et al., 2015) (training $n = 1407$; validation $n = 157$; testing $n = 392$) is applied to 10 LFs defined for these data in the Snorkel documentation.[3] This study employs the original TubeSpam dataset ($n = 1961$) rather than the truncated version used by Snorkel ($n = 1836$). Majority class is 51.28% of data.

**Spouse** This spousal relation information extraction dataset (training $n = 3858$; validation $n = 553$; testing $n = 1101$) is applied to 4 LFs defined for these data in the Snorkel documentation.[4] Majority class is 92.64% of data. This dataset illustrates model performance on a heavily imbalanced dataset, a setting previously explored in programmatic weak supervision (Bringer et al., 2019).

**RNA** This manually labeled dataset consists of PubMed titles obtained for a systematic review on time-course transcriptomics (training $n = 1656$; validation $n = 184$; testing $n = 460$). Titles are labeled according to relevance, with a positive label indicating that a given title is pertinent to the review and a negative label indicating irrelevance. Three LFs were written for this study (Table A.1). Majority class is 62.17% of data. This original dataset is made publicly available under the Open Database License.[5]

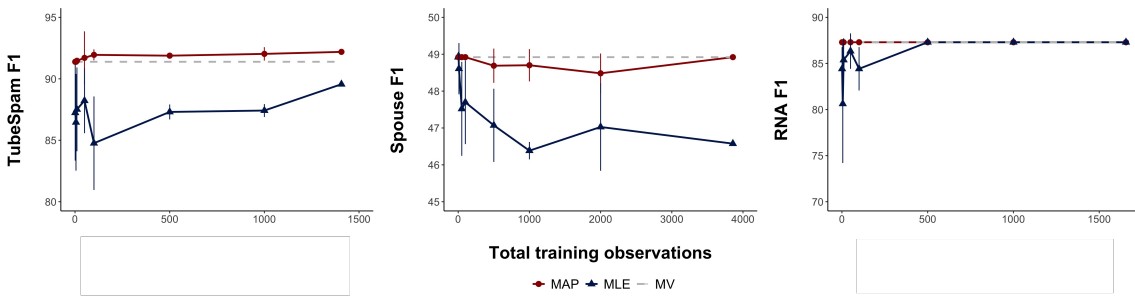

Figure 2: Mean F1 on the full test set for TubeSpam, Spouse, and RNA as training set size increases. Error bars depict standard deviation over 5 replicates. Additional metrics are reported in Figure A.2.

## 4   Results and Discussion

Experimental results highlight three benefits of regularization: 1) priors buoy performance in low-data contexts, 2) MAP meets or exceeds MV and MLE on full data, and 3) model parameters approximate true LF accuracies more effectively than MLE. Reported scores are computed by excluding abstentions, per the convention of Snorkel. Priors over $\alpha$ were derived from MV unless otherwise stated. Performance metrics are reported both for the full test set (Table 2; Figures 2, A.2) and only those instances for which all models voted

---

[3]See https://github.com/snorkel-team/snorkel-tutorials and Snorkel notebooks in https://github.com/regularized-dp/regularized-data-programming/.

[4]Ibid.

[5]https://github.com/regularized-dp/regularized-data-programming/

| | **TUBESPAM** | | | | | | |
|---|---|---|---|---|---|---|---|
| | $MAP_{\alpha^*}$ | $MAP_{mv}$ | MLE | MV | SNO | CAGE | SVM |
| F1 | 94.29 | 92.59 | 89.56 | 91.39 | 88.07 | 70.32 | 94.74 |
| ACCURACY | 94.58 | 93.22 | 89.11 | 92.20 | 87.97 | 62.76 | 94.64 |
| PRECISION | 97.06 | 99.21 | 94.77 | 99.19 | 96.88 | 59.45 | 95.45 |
| RECALL | 91.67 | 86.81 | 84.89 | 84.72 | 80.73 | 86.07 | 94.03 |
| AUC ROC | 94.51 | 93.07 | 89.58 | 92.03 | 88.77 | 62.14 | 94.66 |
| COVERAGE $\hat{y}$ | 75.26 | 75.26 | 89.03 | 75.26 | 89.03 | 100.0 | 100.0 |

| | **SPOUSE** | | | | | | |
|---|---|---|---|---|---|---|---|
| | $MAP_{\alpha^*}$ | $MAP_{mv}$ | MLE | MV | SNO | CAGE | SVM |
| F1 | 48.92 | 48.92 | 46.58 | 48.92 | 46.58 | 12.91 | 18.80 |
| ACCURACY | 65.70 | 65.70 | 64.86 | 65.70 | 64.86 | 15.44 | 91.37 |
| PRECISION | 34.34 | 34.34 | 33.66 | 34.34 | 33.66 | 6.98 | 30.56 |
| RECALL | 85.00 | 85.00 | 75.56 | 85.00 | 75.56 | 85.19 | 13.58 |
| AUC ROC | 73.04 | 73.04 | 68.85 | 73.04 | 68.85 | 47.54 | 55.56 |
| COVERAGE $\hat{y}$ | 18.80 | 18.80 | 20.16 | 18.80 | 20.16 | 100.0 | 100.0 |

| | **RNA** | | | | | | |
|---|---|---|---|---|---|---|---|
| | $MAP_{\alpha^*}$ | $MAP_{mv}$ | MLE | MV | SNO | CAGE | SVM |
| F1 | 87.30 | 87.30 | 87.30 | 87.30 | 82.50 | - | 81.23 |
| ACCURACY | 89.57 | 89.57 | 89.57 | 86.48 | 84.78 | - | 85.43 |
| PRECISION | 80.88 | 80.88 | 80.88 | 80.88 | 73.01 | - | 79.23 |
| RECALL | 94.83 | 94.83 | 94.83 | 94.83 | 94.83 | - | 83.33 |
| AUC ROC | 90.60 | 90.60 | 90.60 | 86.64 | 86.75 | - | 85.02 |
| COVERAGE $\hat{y}$ | 100.0 | 100.0 | 100.0 | 77.17 | 100.0 | - | 100.0 |

Table 2: Comparative performance evaluation on the full test set. Models were trained on the full training set. Testing on all instances in the test set highlights differences in $\hat{y}$ coverage across models, as well as MAP performance gains from forced abstention. MV, $MAP_{mv}$, the $MAP_{\alpha^*}$ benchmark using simulated optimal priors, MLE, CAGE, Snorkel (SNO), and SVM performance scores are expressed as percentages. Dummy accuracy when predicting the majority class is 51.28% of data for TubeSpam, 92.64% for Spouse, and 62.17% for RNA. CAGE could not be run on RNA, as this method requires all LFs to output only a single label (positive or negative) when triggered to vote. Maximum scores are underlined per metric across all DP models, excluding the performance of the supervised SVM.

(Tables 3, A.4). The former represents the real-world use case while the latter enables a direct comparison that excludes the varying abstention patterns of each model.

## 4.1 MAP Aids Performance on Limited Data

MAP was hypothesized to confer performance gains when training instances are limited, with diminishing returns as training set size increases. Training and validation sets were randomly subsetted to simulate a range of data availability ($n = \{1, 5, 10, 50, 100, 500, 1K, 2K, \text{full size}\}$). Performance on the full test set was averaged over five training replicates. MV priors were computed per subset.

MAP generally provided superior performance when data were restricted, while SVM suffered most under limited data (Figures 2, A.2). MLE generally demonstrated higher variance on small training sets, though this instability decreased when evaluating on test instances for which all models vote (Table 3). Spouse provides an example of DP performance for heavily imbalanced data, as negative labels account for 92.64% of observations. MAP minimized false negatives relative to MLE, as exemplified by superior recall for both low- and full-data settings (Table 2; Figures 2, A.2). TubeSpam represents a case where MAP learning quickly exceeds both MLE and MV, with MLE failing to meet MV performance even with increasing training data. Though RNA proved relatively easy to learn for both MAP and MLE, MAP improves mean scores and

| **TUBESPAM** | MAP | MLE | MV | CAGE | SVM |
|---|---|---|---|---|---|
| $n = 500$ | 92.3 (0.7) | 91.8 (0.5) | - | 75.3 (0.0) | 94.4 (0.6) |
| $n = 1407$ | 92.6 | 91.5 | 91.4 | 75.3 | 95.1 |
| **SPOUSE** | MAP | MLE | MV | CAGE | SVM |
| $n = 500$ | 48.9 (0.0) | 48.7 (0.4) | - | 44.6 (0.0) | 12.2 (8.6) |
| $n = 3858$ | 48.9 | 48.9 | 48.9 | 44.6 | 24.6 |
| **RNA** | MAP | MLE | MV | CAGE | SVM |
| $n = 500$ | 87.3 (0.0) | 87.3 (0.0) | - | - | 80.9 (3.3) |
| $n = 1656$ | 87.3 | 87.3 | 87.3 | - | 85.0 |

Table 3: F1 scores on test instances for which all models vote (75.3% for TubeSpam; 18.8% for Spouse; 77.2% for RNA) using $n = 500$ and $n =$ all training instances. Means (standard deviations) are over five replicates. CAGE could not be run on RNA, as this method requires all LFs to output only a single label (positive or negative) when triggered to vote. As MV is not a trained model, we report a single test score for this baseline. Maximum scores are underlined per metric across all DP models, excluding the performance of the supervised SVM.

reduces variance for training sets smaller than 500 observations. As expected, MLE generally approaches MAP performance as training data availability increases.

## 4.2 MAP Meets or Exceeds MLE on Full Data

A fundamental shortcoming of MLE for DP is its potential failure to outperform MV even on full data, as seen with TubeSpam and Spouse when predicting on the full test set (Table 2). Behavior on TubeSpam can be explained by accuracy on instances where the model and MV agree versus disagree (Table A.5). Poor accuracy on the latter showcases the added value of priors over $y$ and forced abstention. As previously observed (Chatterjee et al., 2020), MLE experiences abrupt performance decay, while regularization facilitates a robust training process that is less sensitive to training epoch count (Figure A.3). MLE was more sensitive to hyperparameter values than MAP for TubeSpam and RNA. Conversely, MAP defers to the wisdom of both MV and MLE, ensuring that MAP meets or exceeds the performance of both on all tasks. Unlike MAP, Snorkel and CAGE did not outperform MV on any dataset (Tables 2, 3, A.4). While SVM sometimes exceeds labeling model performance (Table A.4, Figure A.2), this supervised baseline is not a viable method for unlabeled training data. Notably, labeling models outperformed SVM on RNA.

## 4.3 MAP Learns Unknown LF Accuracies

MAP was hypothesized to provide greater fidelity of $\hat{\alpha}$ to true LF accuracies $\alpha^*$. Here model quality is conceptualized as convergence of $\hat{\alpha}$ to $\alpha^*$, measured by the $\ell^2$-norm $||\alpha^* - \hat{\alpha}||_2$. A norm of zero indicates that $\alpha^*$ was learned exactly. We also measure this norm with respect to model priors to demonstrate the contribution of training to this concept of convergence. $\text{MAP}_{mv}$, $\text{MAP}_{\alpha^*}$, and $\text{MAP}_{rn}$ (with randomly selected priors) simulated a range of prior quality.

$\text{MAP}_{mv}$ achieved greater convergence than MLE on two of three datasets, while $\text{MAP}_{\alpha^*}$ achieved greatest convergence on all datasets (Table 4). This suggests that MAP with well-selected priors learns LF accuracies better than MLE. Learning was seen to reduce $||\alpha^* - \hat{\alpha}||_2$ relative to $||\alpha^* - prior||_2$ for $\text{MAP}_{mv}$ in all cases. While diminishing $\ell^2$-norms only improved label quality to a point, stronger convergence to $\alpha^*$ lends greater interpretability to MAP than MLE. Increased regularization through prior strengthening facilitated stronger convergence to prior distribution means. Sufficient prior strength enables recovery of any arbitrary $\hat{\alpha}$.

In contrast, CAGE quality guides failed to provide a natural interpretation for both continuous and discrete LFs (Figure A.4). Though CAGE quality guides are intended to encode prior beliefs over LF accuracies, guide values derived from empirical accuracies and the MV heuristic performed comparably to random values

| | **TUBESPAM** | | **SPOUSE** | | **RNA** | |
|---|---|---|---|---|---|---|
| MODEL | $\|\|\alpha^* - prior\|\|_2$ | $\|\|\alpha^* - \hat{\alpha}\|\|_2$ | $\|\|\alpha^* - prior\|\|_2$ | $\|\|\alpha^* - \hat{\alpha}\|\|_2$ | $\|\|\alpha^* - prior\|\|_2$ | $\|\|\alpha^* - \hat{\alpha}\|\|_2$ |
| $\text{MAP}_{mv}$ | 0.506 | 0.468 | 0.782 | 0.757 | 0.307 | 0.103 |
| $\text{MAP}_{rn}$ | 1.476 | 1.470 | 0.578 | 0.566 | 0.712 | 0.726 |
| $\text{MAP}_{\alpha^*}$ | 0.000 | **0.047** | 0.000 | **0.329** | 0.000 | **0.046** |
| MLE | - | 0.486 | - | 0.512 | - | 0.467 |

Table 4: Convergence of trained model parameters ($\hat{\alpha}$) to true labeling function accuracies ($\alpha^*$) for MLE versus MAP with various priors. We report the distance between $\alpha^*$, prior distribution means, and $\hat{\alpha}$ for MAP with MV priors ($\text{MAP}_{mv}$), MAP with random priors ($\text{MAP}_{rn}$), MAP with empirical accuracy priors ($\text{MAP}_{\alpha^*}$), and MLE, as measured by $\ell^2$-norms. Performance is evaluated for the full training and test sets. Comparing both $\hat{\alpha}$ and the prior to $\alpha^*$ demonstrates the contribution of the learning procedure to whether the parameters converge to the true labeling function accuracies. Minimum values for $\|\|\alpha^* - \hat{\alpha}\|\|_2$ are bolded.

for Spouse and TubeSpam. These results suggest that CAGE offers a less intuitive and less interpretable regularization method than the Bayesian priors introduced in this work.

### 4.4 Limitations and future directions

A fundamental bottleneck to the performance of DP is LF quality, a generally under-formalized and under-studied problem that the present work does not explore (Hsieh et al., 2022). Further, theoretical and empirical evidence suggests that the principled selection of training data subsets can improve performance (Lang et al., 2022b). Combining the proposed regularization method with additional DP extensions such as interactive LF generation (Boecking et al., 2021; Hsieh et al., 2022), LFs that output continuous scores or loss functions (Chatterjee et al., 2020; Sam & Kolter, 2023), and training subset selection (Lang et al., 2022b) could merit further inquiry.

As MV generally offers lower $\hat{y}$ coverage than MLE, improved model performance via forced abstention can come at the price of reduced coverage. Empirical observations on the trade-off between higher quality labels and lower coverage are expressed for the full-data regime in Table 2 and for low-data in Figure A.2.

## 5 Conclusion

The present work introduces a Bayesian extension of DP where MAP replaces MLE as the objective when learning the joint distribution governing a LF matrix. Experimental results suggest that MAP learns LF accuracies more effectively and improves performance in low-data contexts. MV is found to be an effective heuristic for estimating prior parameters. Incorporating MV as a signal at both training and inference time allowed MAP to meet or exceed both MV and MLE performance in all experiments. Results on RNA demonstrate a real-world application for DP in literature curation. Though automated labeling is not recommended under formal guidelines for systematic review (Page et al., 2021a;b), results suggest that judiciously constructed LFs may accelerate title filtering.

## Broader Impact Statement

Data quality is pivotal to the trustworthiness of ML systems, whether data is labeled manually or automatically (Liang et al., 2022). Downstream applications should evaluate label quality uncertainty with respect to user risk in cases where test labels are limited or unavailable, especially in high-stakes contexts such as clinical domains.

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

# A   Appendix

$$\begin{bmatrix} Row & LF0 & LF1 & LF2 & LF3 & LF4 & LF5 & LF6 & LF7 & LF8 & LF9 \\ 0 & 0 & 0 & 0 & 0 & 0 & 1 & 0 & 0 & 0 & 0 \\ 1 & 0 & 0 & 0 & 0 & 0 & 0 & -1 & 0 & 0 & -1 \\ 2 & 1 & 1 & 0 & 0 & 0 & 0 & 0 & 0 & 0 & 0 \\ 3 & 0 & 0 & 0 & 0 & -1 & 0 & 0 & 0 & 0 & 0 \\ 4 & 0 & 0 & 0 & 0 & -1 & 0 & 0 & 0 & 0 & -1 \\ ... & ... & ... & ... & ... & ... & ... & ... & ... & ... & ... \\ 245 & 1 & 1 & 0 & 0 & 0 & 0 & -1 & 0 & 0 & 0 \\ 246 & 0 & 0 & 0 & 0 & -1 & 0 & 0 & 0 & -1 & 0 \\ 247 & 0 & 0 & 0 & 0 & 0 & 1 & 0 & 0 & 0 & 0 \\ 248 & 0 & 0 & 0 & 0 & 0 & 0 & 0 & 0 & 0 & 0 \\ 249 & 0 & 0 & 0 & 0 & -1 & 0 & 0 & 0 & 0 & -1 \end{bmatrix}, \begin{bmatrix} y \\ 1 \\ -1 \\ 1 \\ -1 \\ 1 \\ ... \\ 1 \\ -1 \\ 1 \\ 1 \\ -1 \end{bmatrix}, \begin{bmatrix} \hat{y} \\ 1 \\ -1 \\ 1 \\ -1 \\ -1 \\ ... \\ 1 \\ -1 \\ 1 \\ 0 \\ -1 \end{bmatrix}$$

Figure A.1: **Sample labeling function matrix with ground truth ($y$) and predicted label vectors ($\hat{y}$).** A labeling function matrix may be sparse, with zeros denoting abstention. Zero values are permissible in $\hat{y}$ under the abstention tie-breaking policy, along with class assignments $\in \{-1, 1\}$. Note that $y$ is unavailable at training time and may also be unavailable at test time.

| LABELING FUNCTION | RULE | COVERAGE | CONFLICTS | ACCURACY |
|---|---|---|---|---|
| exclude_organism | -1 if organism is irrelevant, else 0. | 0.05 | 0.04 | 1.00 |
| include_terms | 1 if title contains necessary terms, else -1. | 1.00 | 0.21 | 0.68 |
| exclude_terms | -1 if title contains unwanted terms, else 0. | 0.37 | 0.17 | 1.00 |

Table A.1: **LFs used for transcriptomics literature curation, as designed for this study and applied to the RNA dataset.** A label of -1 denotes that a PubMed title is irrelevant; 1 denotes that the title should proceed to the next stage of systematic review; and 0 denotes abstention. Conflicts are defined as discordance between the output of a given LF and all other LFs. Coverage indicates the percent of observations for which a given LF outputs a label rather than abstaining. Accuracy is measured with respect to the training data by excluding abstained votes (the default behavior of Snorkel). Note that in real-world applications, training labels would be unavailable and LF empirical accuracies would not be calculable.

| | **TUBESPAM** | | **RNA** | |
|---|---|---|---|---|
| METRIC | FIXED EMPIRICAL $\hat{\beta}$ | LEARNED $\hat{\beta}$ | FIXED EMPIRICAL $\hat{\beta}$ | LEARNED $\hat{\beta}$ |
| [TN, FP, FN, TP] | [150, 1, 19, 125] | [148, 3, 17, 127] | [247, 39, 9, 165] | [247, 39, 9, 165] |
| F1 | 92.59 | 92.70 | 87.30 | 87.30 |
| ACCURACY | 93.22 | 93.22 | 89.57 | 89.57 |
| PRECISION | 99.21 | 97.69 | 80.88 | 80.88 |
| RECALL | 86.81 | 88.19 | 94.83 | 94.83 |
| AUC ROC | 93.07 | 93.10 | 90.60 | 90.60 |
| COVERAGE | 75.26 | 75.26 | 1.0 | 1.0 |

Table A.2: **Impacts of learning $\hat{\beta}$ on MAP model performance.** MAP models were trained on the full training set and tested on the full test set, with priors over $\alpha$ chosen using majority vote over the training set. As $\hat{\beta}$ can be directly estimated from the training data, the fixed $\hat{\beta}$ parameters are set to the empirical coverage rates of the labeling functions over the training set. Learned $\hat{\beta}$ parameters were optimized by stochastic gradient descent, with priors based on the empirical coverage rates. The metric [TN, FP, FN, TP] refers to true negatives (TN), false positives (FP), false negatives (FN), and true positives (TP).

For TubeSpam, the $\ell 2$-norm of (learned $\hat{\beta}$ − empirical coverage vector) was 0.00558. For RNA, the $\ell 2$-norm of (learned $\hat{\beta}$ − empirical coverage vector) was 0.00289. As the learned $\hat{\beta}$ converges strongly to the empirical coverage vector and performance metrics were unchanged or minimally affected by learning, we opted for model simplicity by employing the fixed $\hat{\beta}$ for subsequent experiments.

| **MAP$_{mv}$** | | SELECTED VALUES | | |
|---|---|---|---|---|
| HYPERPARAMETER | SEARCH VALUES | TUBESPAM | RNA | SPOUSE |
| Prior strength scaling factor | $\{10, 100\}$ | 10 | 10 | 10 |
| Learning rate | $\{0.001, 0.01\}$ | 0.01 | 0.01 | 0.01 |
| Initialization value for all $\alpha_j$ | $\{0.8, 0.9, 1.0\}$ | 1.0 | 1.0 | 1.0 |
| Parameter $p$ for prior over $y$ | $[0.0, 0.1, ..., 0.9, 1.0]$ | 0.5 | 0.5 | 0.5 |
| Force abstention | $\{$True, False$\}$ | True | False | True |

| **MAP$_{\alpha^*}$ BENCHMARK** | | SELECTED VALUES | | |
|---|---|---|---|---|
| HYPERPARAMETER | SEARCH VALUES | TUBESPAM | RNA | SPOUSE |
| Prior strength scaling factor | $\{10, 100\}$ | 10 | 10 | 10 |
| Learning rate | $\{0.001, 0.01\}$ | 0.01 | 0.01 | 0.01 |
| Initialization value for all $\alpha_j$ | $\{0.8, 0.9, 1.0\}$ | 1.0 | 1.0 | 1.0 |
| Parameter $p$ for prior over $y$ | $[0.0, 0.1, ..., 0.9, 1.0]$ | 0.5 | 0.5 | 0.7 |
| Force abstention | $\{$True, False$\}$ | True | False | True |

| **MLE** | | SELECTED VALUES | | |
|---|---|---|---|---|
| HYPERPARAMETER | SEARCH VALUES | TUBESPAM | RNA | SPOUSE |
| Learning rate | $\{0.001, 0.01\}$ | 0.001 | 0.001 | 0.01 |
| Initialization value for all $\alpha_j$ | $\{0.8, 0.9, 1.0\}$ | 0.9 | 0.8 | 1.0 |

| **SNORKEL** | | SELECTED VALUES | | |
|---|---|---|---|---|
| HYPERPARAMETER | SEARCH VALUES | TUBESPAM | RNA | SPOUSE |
| Training epochs | $\{50, 100, 250\}$ | 50 | 50 | 100 |
| Learning rate | $\{0.001, 0.01, 0.1\}$ | 0.1 | 0.01 | 0.01 |
| $\ell 2$ regularization strength | $\{0.0, 0.2, 0.4\}$ | 0.2 | 0.4 | 0.0 |

| **CAGE** | | SELECTED VALUES | | |
|---|---|---|---|---|
| HYPERPARAMETER | SEARCH VALUES | TUBESPAM | RNA | SPOUSE |
| Learning rate | $\{0.01, 0.001\}$ | 0.001 | - | 0.01 |
| Guide value for all LFs | $[0.4, 0.5, ..., 0.8, 0.9]$ | 0.4 | - | 0.4 |
| Initialization value for all $\theta_j$ | $\{0.8, 0.9, 1.0\}$ | 0.9 | - | 0.8 |

| **SUPPORT VECTOR MACHINE** | | SELECTED VALUES | | |
|---|---|---|---|---|
| HYPERPARAMETER | SEARCH VALUES | TUBESPAM | RNA | SPOUSE |
| $\ell 2$ regularization strength | $\{0.1, 0.01, 0.001, 0.0001\}$ | 0.001 | 0.0001 | 0.0001 |

Table A.3: **Hyperparameter values combinatorially explored through grid search.** Performance was evaluated for the full training set. Selected value combinations were those that yielded the highest scores for the largest number of performance metrics on the validation split of each dataset (among accuracy, F1 score, precision, recall, and AUC ROC). MLE was found to be more sensitive to hyperparameter values than MAP for TubeSpam and RNA. Note that optimal hyperparameter combinations were rarely unique; in this case, model training time and simplicity were prioritized. All models used vanilla SGD except CAGE, which used the Adam optimizer as employed in the original paper. MAP and MLE models employed early stopping using the validation loss with a patience of 5 epochs. MAP$_{mv}$ was trained with priors derived from majority vote over the training data. MAP$_{\alpha^*}$ used simulated optimal priors derived from the empirical LF accuracies with respect to the training data, as computed by the Snorkel Python package. MAP$_{\alpha^*}$ serves as a benchmark for the evaluation of MAP$_{mv}$, providing an experimental control to demonstrate the utility of the majority vote heuristic for automated prior selection (see Tables 2, A.4).

| | **TUBESPAM** | | | | | | |
|---|---|---|---|---|---|---|---|
| | $\text{MAP}_{\alpha*}$ | $\text{MAP}_{mv}$ | MLE | MV | SNO | CAGE | SVM |
| F1 | 94.29 | 92.59 | 91.51 | 91.39 | 91.39 | 75.28 | 95.07 |
| ACCURACY | 94.58 | 93.22 | 92.20 | 92.20 | 92.20 | 70.17 | 95.25 |
| PRECISION | 97.06 | 99.21 | 97.64 | 99.19 | 99.19 | 63.21 | 96.43 |
| RECALL | 91.67 | 86.81 | 86.11 | 84.72 | 84.72 | 93.06 | 93.75 |
| AUC ROC | 94.51 | 93.07 | 92.06 | 92.03 | 92.03 | 70.70 | 95.22 |
| COVERAGE $\hat{y}$ | 75.26 | 75.26 | 75.26 | 75.26 | 75.26 | 75.26 | 75.26 |
| | **SPOUSE** | | | | | | |
| | $\text{MAP}_{\alpha*}$ | $\text{MAP}_{mv}$ | MLE | MV | SNO | CAGE | SVM |
| F1 | 48.92 | 48.92 | 48.92 | 48.92 | 48.92 | 44.59 | 24.56 |
| ACCURACY | 65.70 | 65.70 | 65.70 | 65.70 | 65.70 | 60.39 | 79.23 |
| PRECISION | 34.34 | 34.34 | 34.34 | 34.34 | 34.34 | 30.56 | 41.18 |
| RECALL | 85.00 | 85.00 | 85.00 | 85.00 | 85.00 | 82.50 | 17.50 |
| AUC ROC | 73.04 | 73.04 | 73.04 | 73.04 | 73.04 | 68.79 | 55.76 |
| COVERAGE $\hat{y}$ | 18.80 | 18.80 | 18.80 | 18.80 | 18.80 | 18.80 | 18.80 |
| | **RNA** | | | | | | |
| | $\text{MAP}_{\alpha*}$ | $\text{MAP}_{mv}$ | MLE | MV | SNO | CAGE | SVM |
| F1 | 87.30 | 87.30 | 87.30 | 87.30 | 87.30 | - | 85.04 |
| ACCURACY | 86.48 | 86.48 | 86.48 | 86.48 | 86.48 | - | 85.63 |
| PRECISION | 80.88 | 80.88 | 80.88 | 80.88 | 80.88 | - | 86.83 |
| RECALL | 94.83 | 94.83 | 94.83 | 94.83 | 94.83 | - | 83.33 |
| AUC ROC | 86.64 | 86.64 | 86.64 | 86.64 | 86.64 | - | 85.59 |
| COVERAGE $\hat{y}$ | 77.17 | 77.17 | 77.17 | 77.17 | 77.17 | - | 77.17 |

Table A.4: **Comparative performance evaluation on test set instances for which all models vote.** Models were trained on the full training set. As all models are evaluated on an identical subset of the test data, $\hat{y}$ coverage is identical. MV, $\text{MAP}_{mv}$, the $\text{MAP}_{\alpha*}$ benchmark using simulated optimal priors, MLE, CAGE, Snorkel (SNO), and SVM performance scores are expressed as percentages. Dummy accuracy when predicting the majority class is 51.28% of data for TubeSpam, 92.64% for Spouse, and 62.17% for RNA. CAGE could not be run on RNA, as this method requires all LFs to output only a single label (positive or negative) when triggered to vote. Maximum scores are underlined per metric across all DP models, excluding the performance of the supervised SVM.

| | **TUBESPAM** | | **RNA** | |
|---|---|---|---|---|
| | MV-CONCORDANT | MV-DISCORDANT | MV-CONCORDANT | MV-DISCORDANT |
| **MAP MODEL** | | | | |
| ACCURACY (%) | 93.15 | 64.91 | 86.48 | 100.00 |
| MV ABSTENTIONS (%) | 12.84 | 94.74 | 0.0 | 100.00 |
| **MLE MODEL** | | | | |
| ACCURACY (%) | 94.41 | 65.07 | 86.48 | 100.00 |
| MV ABSTENTIONS (%) | 13.07 | 85.71 | 0.0 | 100.00 |

Table A.5: **Disaggregating performance by concordance with majority vote (MV).** Performance is evaluated for the full training and test sets. Accuracy when predicting on instances for which a labeling model and MV agree (MV-concordant) versus disagree (MV-discordant) explain the benefit of forced abstention on TubeSpam. The majority of discordant predictions were MV abstentions for both datasets. MAP trained on RNA does not benefit from forced abstention, as the model achieves 100% accuracy on MV-discordant instances, all of which were instances for which MV abstained.

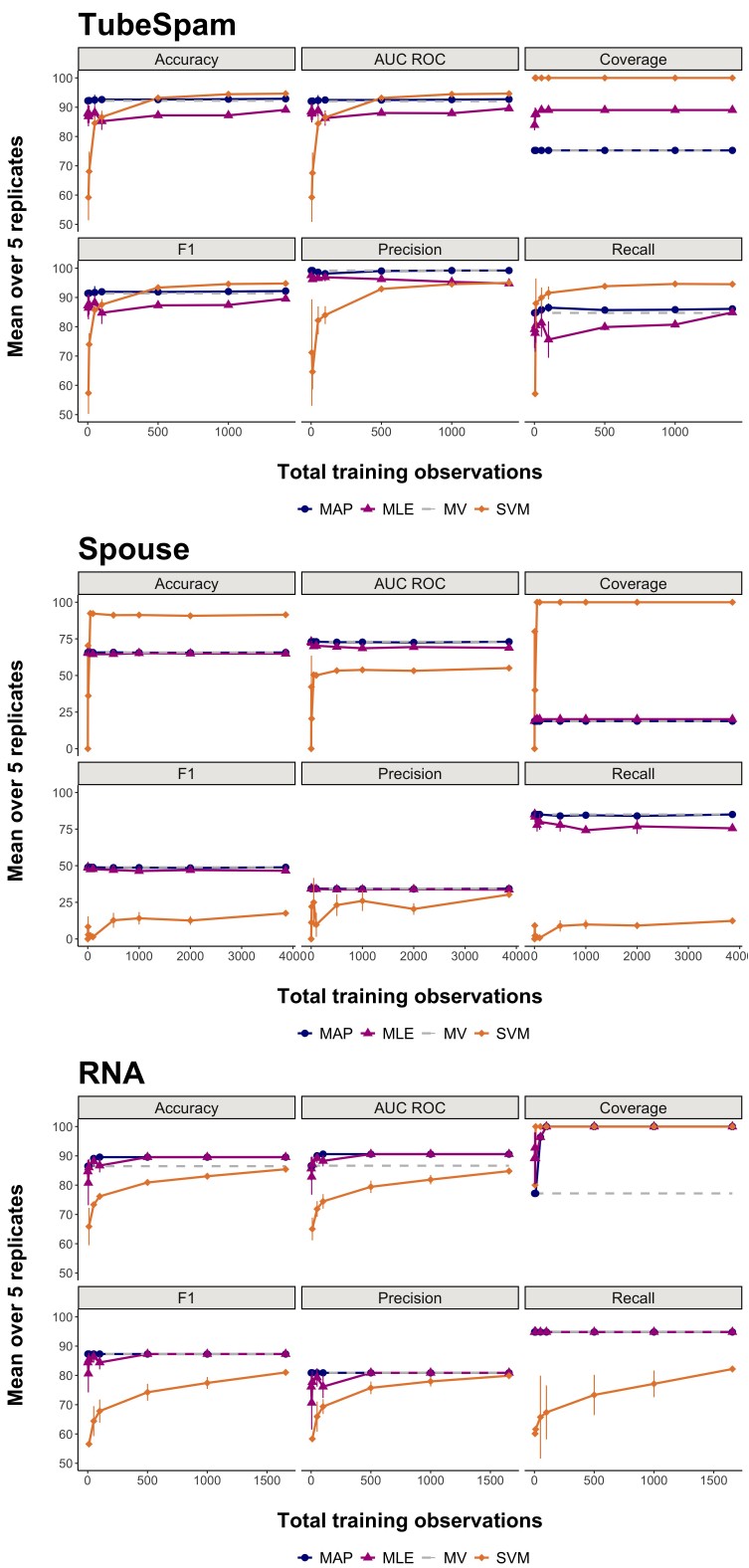

Figure A.2: **Low-data regimes for TubeSpam (top), Spouse (center), and RNA (bottom).** Performance of MAP with MV priors, MLE, MV, and SVM as training set size increases. Performance is evaluated for the full test set. Means are across five random splits and error bars depict standard deviation.

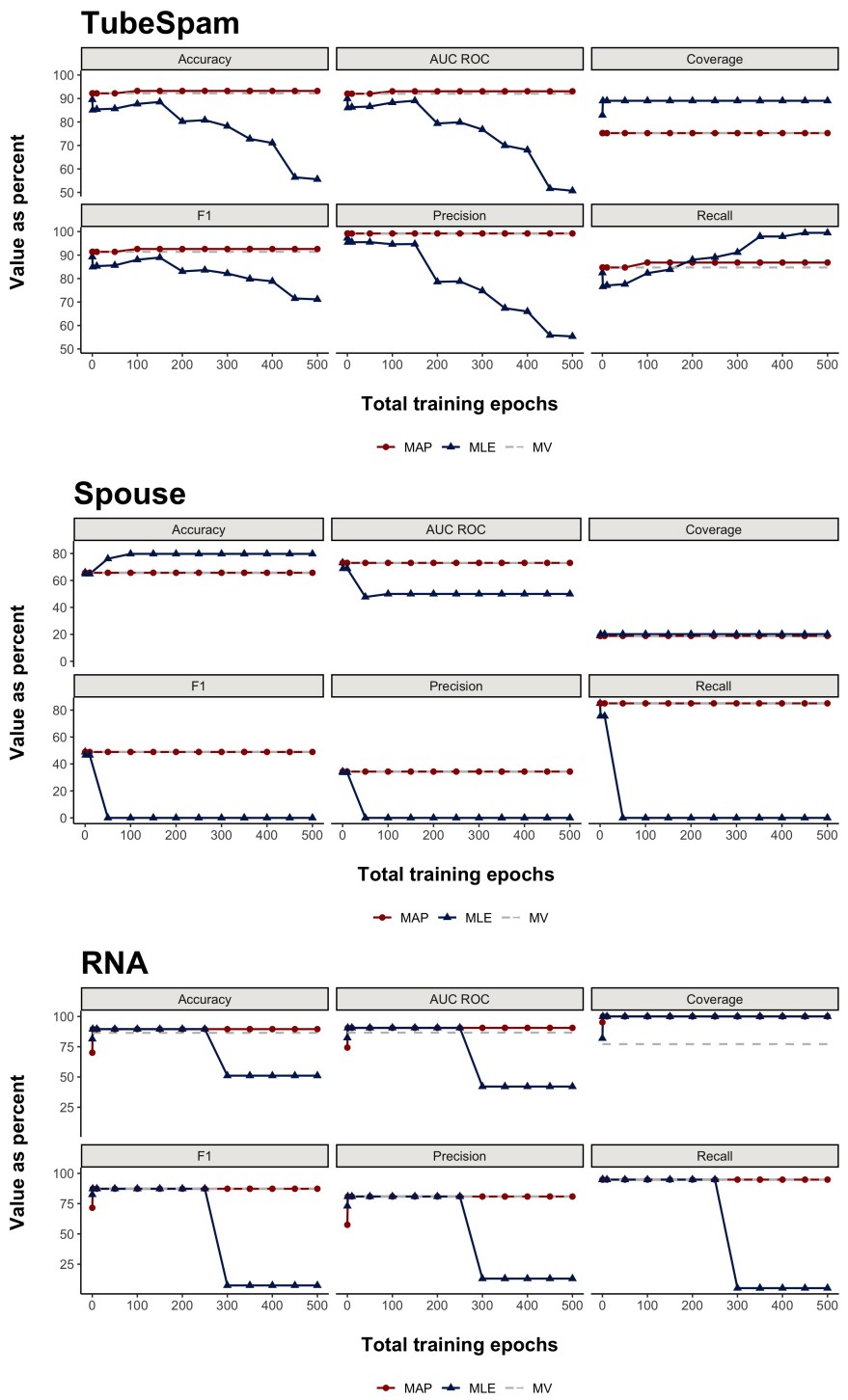

Figure A.3: **Performance stability on TubeSpam (top), Spouse (center), and RNA (bottom).** MAP and MLE performance stability as training epochs increase. Performance is evaluated for the full training and test sets. Majority vote (MV) represents baseline performance. Similar trends were observed for regularization versus MLE by Chatterjee et al. 2020 in their comparison of CAGE and Snorkel.

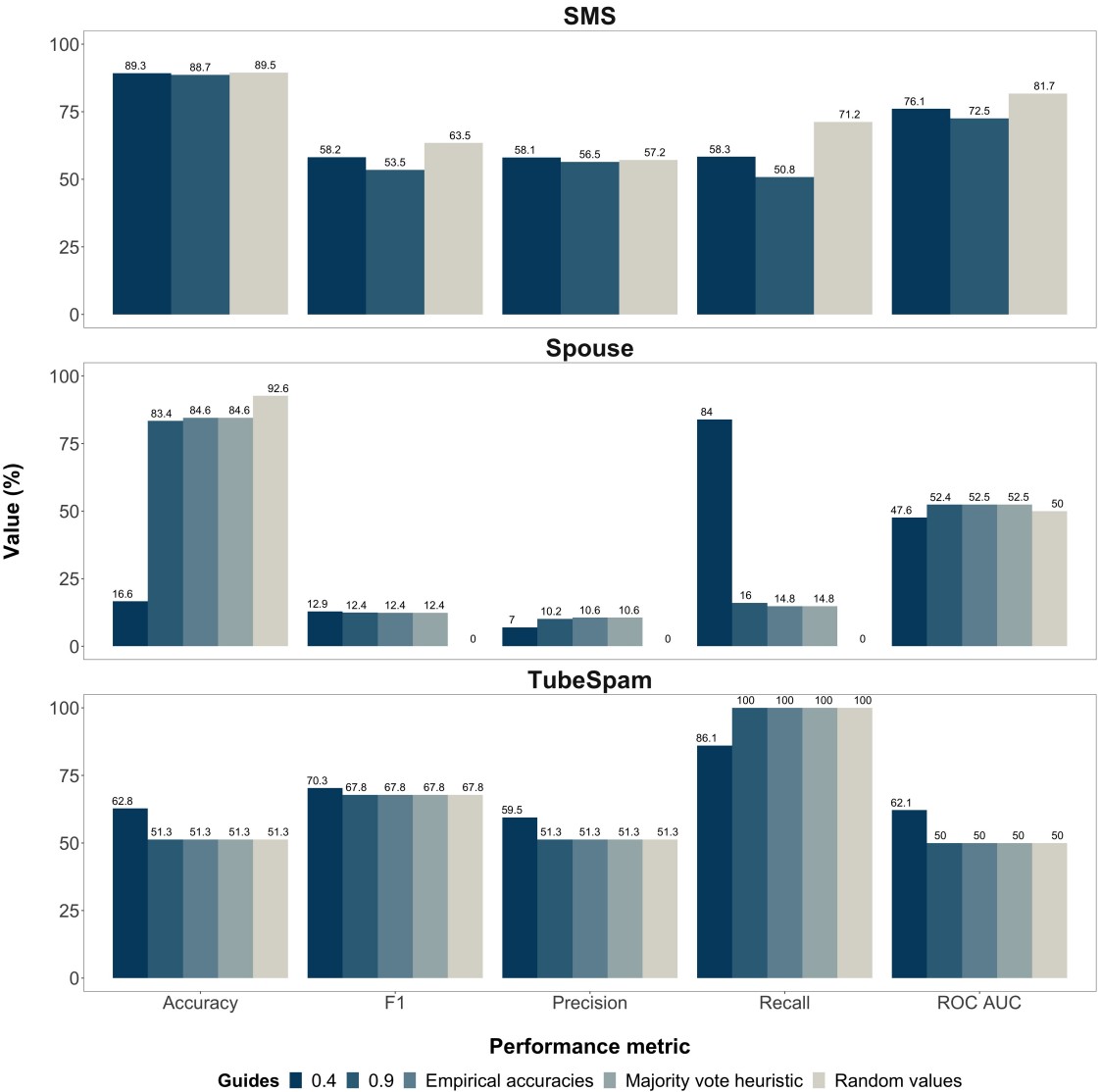

Figure A.4: **CAGE performance with varying guide values.** Guides were set to **1)** the values used in the original publication (Chatterjee et al., 2020), which assigns a guide value of 0.9 to all LFs; **2)** a randomly selected low value uniformly assigned to all LFs (0.4); **3)** random values between 0 and 1 that vary per LF; **4)** the empirical accuracies of each respective LF on the training data; and **5)** the majority vote heuristic used to derive the Bayesian priors presented in this paper. For SMS, CAGE model hyperparameters and the SMS dataset are exactly as provided by the authors (https://github.com/oishik75/CAGE), with the exception of guide values. The continuous LF loss function was used for SMS, while the discrete LF loss was employed for TubeSpam and Spouse.

For the SMS dataset, both methods of random guide selection (0.4 and random values) outperformed the guides reported in Chatterjee et al. 2020 for all performance metrics. Empirical accuracies were unavailable for SMS training data. For TubeSpam, uniformly assigning guide values of 0.4 performed best, despite LF empirical accuracies ranging from 59.4% to 100%. Similarly, uniform guide values of 0.4 performed best for Spouse on the basis of F1 and recall (the metrics of interest, as 92.6% of true labels were negative) despite LF empirical accuracies ranging from 39.9% to 96.8%. Though CAGE guides are intended to encode prior beliefs over LF accuracies, guide values derived from empirical accuracies and the majority vote heuristic performed equivalently to non-uniform random values for TubeSpam. These results suggest that CAGE guides offer a less intuitive and less natural interpretation than the Bayesian priors introduced in this work.

