# OpenReview forum: "Regularized Data Programming with Automated Bayesian Prior Selection"
_TMLR — Withdrawn by Authors_

### Review · Reviewer_ghMW · 2023-07-30

**Summary Of Contributions:**

This paper introduces a Bayesian extension of data programming (DP), offering a change in the conventional learning objective from maximum likelihood estimation (MLE) to maximum a posteriori (MAP). The researchers propose this adjustment to the parameter estimation, using a Bernoulli prior over the latent label and a Beta prior over the labeling function accuracies. This is done with an intent to regularize estimation, particularly in settings with limited data availability.

**Audience:**

Yes

**Broader Impact Concerns:**

Nothing to note here.

**Claims And Evidence:**

Yes

**Requested Changes:**

Typically in similar studies, a supervised approach serves as the hypothetical performance upper-bound. Here, however, the supervised SVM performed excellently in some experiments but not in others. While it is expected that a supervised approach doesn't work well with limited training data, it is maybe fair to ask why such an approach is used at all when there is little training data ("unlabeled training data" as the paper puts it). Is there any alternative for the readers to gauge the performance upper-bound?

**Strengths And Weaknesses:**

The authors' incorporation of majority voting as an unsupervised estimator for prior parameters is clever and technically sound, drawing upon well-established methods for parameter estimation. Empirical results showcase the superiority of the proposed MAP formulation over the methods including conventional MLE and CAGE, especially in low-data scenarios. These findings are also competitive with supervised baselines such as SVM.

However, the paper’s novelty lies primarily in its successful utilization of majority voting as a proxy signal to estimate the parameters of the prior, rather than in the general formulation itself. It might be a new set of experiments in the DP research, but the observation that replacing MLE with MAP works better in a low-data situation is hardly a surprising one.

In conclusion, this paper’s MAP formulation offers promising potential for addition to existing DP frameworks, especially in bridging the performance gap in low-data scenarios. Despite questions regarding some technical details, the novelty of the overall approach and its impact, the core idea is clear and well-executed. It may inspire further experimentation in research and applications.

---

### Review · Reviewer_7KMZ · 2023-09-11

**Summary Of Contributions:**

The paper presents a Bayesian treatment of Data Programming (DP), replacing Maximum Likelihood Estimation (MLE) with Maximum A Posteriori (MAP) estimation as the objective for learning the joint distribution governing Labeling Functions (LF). The experimental results provides some evidence that this Bayesian extension may effectively learn LF accuracies and leads to improved performance, particularly in low-data scenarios. Additionally, the authors propose some practical ideas about incorporating Majority Vote (MV) as a signal for estimating prior parameters. The introduction of a new DP benchmark dataset is a noteworthy contribution.

**Audience:**

Yes

**Broader Impact Concerns:**

The paper appears to address a problem analogous to labeling in crowdsourcing, as initially introduced by Raykar et al. in 2010 (Learning From Crowds, JMLR 2010). While Raykar et al. focused on Bayesian priors on experts, this work extends the concept to labeling functions. The paper could provide additional context regarding how it builds upon or diverges from prior work in this field to highlight its unique contributions.

**Claims And Evidence:**

No

**Requested Changes:**

Abstention Cases: The exclusion of abstention cases from empirical analysis raises questions about the model's treatment of abstention, which is a particularly interesting aspect of the model. The paper could benefit from a thorough exploration of modeling abstention, perhaps by defining Bayesian priors on the abstention rate of each labeling function. This could lead to a more comprehensive understanding of the model's performance, especially when dealing with tasks involving a high rate of abstention.

Clarification of Parameter "p": The paper should provide a more detailed explanation of the parameter "p" applied to Bernoulli priors over "y." Readers may need clarification on the significance of "p" and its connection to majority voting. Additionally, the sentence, "Wherever y_mv votes, we set p ≥ 0.5 such that sufficiently strong priors (p approaching 1) will force the model to recapitulate every vote in y_mv," could be further elaborated to clarify how this relates to majority voting and its impact on model behavior.

Significance of MAP Estimates: The paper should elaborate on why Maximum A Posteriori (MAP) estimates are chosen over Maximum Likelihood (ML) estimates in this specific context. Exploring the significance of MAP estimates, particularly in relation to class imbalance or abstention in voting, would provide a clearer understanding of the model's design choices.

Figure 1 Caption Typo: A typographical error in the caption of Figure 1, where "unavailabe" is used, should be corrected for clarity.

Section 1.1: In Section 1.1, there are spelling errors such as "developement of achemical reaction database" and "systamatic reviews," which should be corrected for accuracy.

**Strengths And Weaknesses:**

Strengths:

Bayesian Extension: The incorporation of a Bayesian extension facilitates the integration of prior knowledge into the model and guides the labeling tendencies of labeling functions.

Benchmark Dataset: The introduction of a new DP benchmark dataset for biomedical literature curation is a notable strength, enhancing the quality and standardization of research in this domain.

Weaknesses:

Limited Empirical Evaluation: While the paper mentions an application in biomedical literature curation, the empirical evaluation is somewhat constrained. Providing more concrete examples or case studies to showcase the practical utility of the proposed Bayesian DP approach in real-world scenarios would enhance the paper's impact. Notably, in the biomedical literature curation datasets, both Maximum Likelihood Estimation (MLE) and Maximum A Posteriori (MAP) methods perform comparably. The only instance where the proposed method slightly outperforms MLE is in the TUBESPAM data, which may not be as relevant as real-world datasets.

Limited Technical Novelty: The paper introduces a Bayesian extension of DP, incorporating Bayesian priors on true class labels and labeling function accuracy. While this is an important contribution, the paper relies on limited experimental evidence to favor the proposed approach. A more robust justification of the need for such a Bayesian approach, supported by additional theoretical and empirical insights, would strengthen the paper's argument.

Clarity of Explanations: Some sections of the paper, particularly those explaining the technical details of the Bayesian extension, may be challenging to comprehend, impacting the overall clarity of the paper.

---

### Review · Reviewer_CJkt · 2023-10-30

**Summary Of Contributions:**

The authors consider the problem of determining labels for instances as combination of a collection of weak labelling functions. The authors add a standard conjugate prior for a previously used probabilistic model and consequently change from maximum likelihood to MAP estimation of the model parameters. The method is compared against previous methods on a collection of small benchmark data sets and the method is shown to have in general comparable performance; it helps a bit when labelling only a small number of samples as we should expect from a regularised model but otherwise works the same as ML estimate of the same model.

**Audience:**

No

**Broader Impact Concerns:**

There is a broader impact statement but it is a bit lacking. The authors acknowledge that there are risks in using automated labelling protocols and even cite works that recommend not using them, yet they propose a concrete algorithm and open software for doing it in cases where the general theoretical argumentation behind the algorithm family does not seem to hold. This would warrant stating the risks of this specific method more clearly, instead of just referring to data quality being pivotal.

**Claims And Evidence:**

No

**Requested Changes:**

The authors would need to motivate properly why we used use DP in small sample cases in the first place. Ideally this should be done by providing theoretical guarantees that justify the use of the approach in limited data scenarios, but most likely the proposed algorithms does not have such guarantees. This means the discussion should be broadened considerably and the method should be presented also in context of alternative solutions for the problem. For instance, what do we give up if relying on DP in this regime compared to completely alternative ways of labelling the data.

The presentation in Section 2 should be improved so that all of the details are presented in sufficient level of formality; now Section 2.3 seems to describe important elements of the solution but it is hard to follow what exactly is stated.

**Strengths And Weaknesses:**

The basic approach is sound but extremely straightforward. In effect, the authors simply plug in a prior/regularizer for a specific learning objective and motivate the change by the standard argumentation, primarily improvements in the small-sample regime. Since standard gradient descent is used for learning the MAP estimate, just as in the previous work considering the ML estimate, no new algorithm development was required. Both the mathematical development and algorithm implementation are approximately on a level that could be had as an exercise in a bachelor level course. Considering proper Bayesian estimates that also quantify the uncertainty would perhaps been a bit more interesting, and could be done with similar out-of-the-box algorithms.

Even though there is no scientific novelty in the work, it could still be interesting for people working on DP. However, from this perspective the paper is problematic as it deviates from the common use cases and lacks theoretical analysis. Rather et al. (2016) motivates the original method from the perspective of creating *large* training data sets and conducts all experimentation on cases with hundreds of thousands or millions of generated labels. In this paper the experimentation is done in cases where the sizes are hundreds or thousands. At least I personally would not turn to this family of methods in applications like this, but would rather consider e.g. active learning strategies or more complex probabilistic models for aggregating the labels as computational cost is not going to be an issue.

I would also expect people interested in DP to care about the theoretical analysis of the algorithm, but the paper does not provide any additional analysis or even confirm that the previous theoretical guarantees still hold. For an automatic process it is important to have strong guarantees, as also pointed out implicitly by the authors who cite Page et al. (2021) that recommends not using automated labelling at all in certain applications. If I am not mistaken, DP addresses these challenges by guaranteeing convergence at the limit of large data sets (or for large number of labelling functions, but they do show that O(1) functions are enough), and by stepping out of that regime you lose the guarantees. In other words, we cannot rely any more on the theoretical basis of DP and hence should provide new justification.

The paper is in general relatively easy to follow, but the presentation still leaves a bit to desire. When entering the small sample regime the work should be put into a broader perspective, describing the advantages and disadvantages of the proposed method compared to alternative ways of labelling the samples (e.g. active learning, or even comprehensive manual annotation that is often feasible on this scale) in addition to previous work on DP. The mathematical notation in Section 2.1 is unconventional as Eq. (1) is written in form of if-clauses that are conditioned on the left-hand-side of the probability, and Section 2.3 is difficult to understand. I would expect formal mathematical presentation also in that section.

The experimentation appears sound, but is somewhat limited, showing only how the method works on three small data sets. The method works as expected, showing some improvement over MLE when limited to very few samples but e.g. for the RNA data there is no real difference.

Overall, I do not consider this as ready for publication and the paper falls short of the TMLR evaluation criteria. It is not likely to be interesting for people working on DP as it lacks theoretical guarantees, it is not interesting for probabilistic modelling people as the technical contribution is simply plugging in standard conjugate prior, and there is no empirical evidence that would suggest the method to be highly valuable in practice. The paper does not make particularly clear claims, but instead explains contributions on the level of "we introduce ..." both in Introduction and Conclusions, limiting summary of the results also on the level of "demonstrate an application" and "meet of exceed MLE performance". In other words, the authors do not even make scientific claims but simply list what they did and observed.

---

### Note · Authors · 2023-11-08

**Comment:**

We sincerely thank the reviewers for their thorough and insightful feedback. Unfortunately, we are unable to incorporate review due to unforeseen circumstances. We hope that the record of this preprint and peer review might be useful for future work in this research area.

**Withdrawal Confirmation:**

I have read and agree with the venue's withdrawal policy on behalf of myself and my co-authors.